# Parental Child-Feeding in the Context of Child Temperament and Appetitive Traits: Evidence for a Biopsychosocial Process Model of Appetite Self-Regulation and Weight Status

**DOI:** 10.3390/nu12113353

**Published:** 2020-10-30

**Authors:** Jeffrey Liew, Zhiqing Zhou, Marisol Perez, Myeongsun Yoon, Mirim Kim

**Affiliations:** 1Department of Educational Psychology, Texas A & M University, College Station, TX 77843, USA; zhou_zq@tamu.edu (Z.Z.); myoon@tamu.edu (M.Y.); mrKim37@tamu.edu (M.K.); 2Department of Psychology, Arizona State University, Tempe, AZ 85281, USA; marisol.perez@asu.edu

**Keywords:** temperament, appetitive traits, parental feeding, childhood obesity

## Abstract

Pediatric obesity is a serious public health challenge and there is a need for research that synthesizes the various linkages among the child and parental factors that contribute to pediatric overweight and obesity. The main objective of this study was to examine potential mechanisms and pathways that might explain how child temperament is indirectly related to child body composition through appetitive traits and parental child-feeding practices. Participants consisted of 221 children between 4–6 years of age (51% males, mean age = 4.80 years, standard deviation = 0.85) and their parents (90.5% biological mothers, (Mage) = 32.02 years, (SDage) = 6.43) with 71% of the parents being married. Study variables included child temperament (negative affectivity and effortful control), child appetitive traits (food avoidance and food approach), controlling parental child-feeding practices (restrictive feeding and pressure to eat), and child body composition. Body composition were indexed by parent perceptions, body mass index (BMI), and percent body fat. Results showed that children with low levels of effortful control are more prone to exhibit food avoidance, which in turn is likely to elicit parental pressure to eat that in turn is linked to high child weight status. In addition, children with high levels of negative affectivity are prone to exhibit a food approach, which in turn is likely to elicit restrictive feeding from parents that in turn is linked to high child objective weight status. Findings situate controlling parental child-feeding practices in the context of child temperament and appetitive traits using a biopsychosocial framework of appetite self-regulation and weight. Results highlight that child appetite self-regulation processes and parental child-feeding practices could be essential components to target in childhood obesity preventive interventions.

## 1. Introduction

Pediatric obesity has been recognized as one of the most serious public health challenges in the 21st century, because it is associated with substantial morbidity and mortality and often leads to significant financial, medical, and quality of life impacts [1,2]. The prevalence of childhood obesity has been high or rising in low-, middle-, and high-income countries throughout North America, Eastern Europe, Pacific island nations, and the Middle East, pointing to the need for research focused on identifying modifiable risk factors and pathways that leverage on the expertise across disciplines such as the developmental science and obesity literature. Early child temperament has been suggested as a factor that partly influences why some children are more likely than others to exhibit low levels of appetite self-regulation that over time could contribute to childhood overweight or obesity [3,4].

To date, very few studies have examined relations between children’s temperament and appetitive traits and their unique or joint contributions to weight status [3,5]. Importantly, despite the temperamental basis for emotional self-regulation and emotional or impulsive eating, the behaviors associated with such traits and parents’ responses to such traits and behaviors are modifiable through intervention [6]. According to a biopsychosocial model of the development of children’s overweight and obesity [7], child biological foundations (e.g., child temperament and child appetitive traits) are assumed to influence parents’ cognitions, expectations, and interpretations in addition to parents’ reactions or behaviors (e.g., parents feeding styles and practices). Guided by a biopsychosocial framework, the present study examines the contributions of child temperament (negative affectivity and effortful control), child appetitive traits (food approach and food avoidance), and parental controlling child-feeding behaviors (restrictive feeding and pressure to eat) to child body composition (Body Mass Index or BMI, percent body fat, and parental perception of child weight) in an economically and ethnically diverse sample of 4–6-year-old children. We focus on this age group (4–6 years) for this study, because abilities for emotional self-regulation that involve executive or top-down control (including effortful control) begin to consolidate after the first 3 years of life [8,9].

### 1.1. Child Temperament and Appetitive Traits

Temperament refers to individual differences in reactivity and self-regulation that are evident from infancy [8,9]. Child temperament is biologically based [10] and is considered the early foundation of later personality and long-term styles of behaving and reacting to the environment [11]. Negative affectivity and effortful control are two core temperament factors important for emotional self-regulation (also referred to as emotion regulation in the literature), with negative affectivity representing reactive or reflexive (e.g., bottom-up) processes and effortful control representing control or regulatory (e.g., top-down) processes. Negative affectivity is the disposition to experience aversive emotional states (e.g., anger, discomfort, fear, sadness) without being able to be easily soothed [9]. Effortful control is defined as the ability to inhibit a dominant response in order to perform a sub-dominant response [9]. For example, when a child who really loves chocolates gets a box of chocolates and is able to eat one piece, stop, and put the box away to save the rest to eat later, that child is exerting and demonstrating effortful control.

One way that personality has been represented is by approach and avoidance temperaments, and this framework has been extended to food behaviors. Approach temperament is a general neurobiological sensitivity to real or imagined positive stimuli or rewards, while avoidance temperament is a general neurobiological sensitivity to real or imagined negative stimuli or punishments [12]. When the stimuli are foods, children’s appetite self-regulation [13,14] could also be broadly represented by two appetitive traits: food approach and food avoidance [15]. Studies suggest that children’s temperament traits important for emotional self-regulation are associated with food approach and food avoidance. Negative affectivity has been linked to food avoidance, while poor self-regulation (including impulsivity) has been linked to food approach [4,16,17]. Thus, while emotional self-regulation and appetite self-regulation processes are distinct, they are also related to one another.

Prior research has shown that child appetitive traits are associated with pediatric obesity [18]. Child appetitive traits may impact child eating behavior, which may then influence their weight status [19]. Specifically, food approach tendencies (e.g., faster eating rates, greater responsiveness to food-cues, emotional overeating) have been linked to higher weight status, while food avoidance tendencies (e.g., food fussiness, slower eating rate) have been linked to lower weight status [18,20].

### 1.2. Parental Controlling Child-Feeding Behavior

Young children are dependent on their caregivers for food intake and nutrition. Thus, parental child-feeding behaviors are considered one of the most proximal mechanisms for children’s dietary intake [21]. When parents engage in nonresponsive or coercive and intrusive feeding behaviors, children have fewer opportunities to learn and develop self-regulation of eating [22,23]. Some common parental controlling feeding behaviors include pressure to eat, which happens when parents pressure their children to eat more or to eat certain foods, and restrictive feeding, which happens when parents restrict their children’s food consumption, particularly calorie-dense foods [24]. Previous research found that pressure to eat was correlated with lower child weight status while restrictive feeding was correlated with higher child weight status [18,25,26,27].

Parental child-feeding practices could influence the amount and type of foods that children eat, but children’s appetite self-regulation processes could also influence parents’ feeding practices. For example, results from longitudinal studies suggested that parental controlling feeding (e.g., pressure to eat and restrictive feeding for weight-control purposes) was prospectively related to children’s food approach and tendency to overeat [28], and child satiety responsiveness at 2 years (i.e., fullness sensitivity indicative of good self-regulation of eating) is prospectively related to maternal feeding practices (covert and overt restrictive feeding 3 years later) [29]. Overall, there appears to be a bi-directional relationship between child appetite self-regulation processes and parental child-feeding behaviors [30,31].

### 1.3. Present Study

A growing body of research has documented separate associations among child temperament and appetitive traits, controlling parental child-feeding practices, and pediatric overweight and obesity. However, there is a need for research that synthesizes the various linkages among the child and parental factors that contribute to pediatric overweight and obesity.

The main objective of this study was to examine potential mechanisms and pathways that might explain how child temperament is indirectly related to child body composition through child appetitive traits and parental child-feeding practices. Specifically, we test a proposed model that is shown in Figure 1. Given the genetic basis of temperament [10], we included negative affectivity and effortful control as the exogeneous variables in the model. We conceptualized child appetitive traits (food approach and food avoidance) as expressions of approach and avoidance temperaments in relation to food, and thus are influenced by negative affectivity and effortful control; these temperament and appetitive traits contribute to the process of appetite self-regulation. While we acknowledge the bidirectional relation between child appetitive traits or appetite self-regulation and parental child-feeding behaviors, for the purpose of the present study, we conceptualized parental controlling feeding behaviors (restriction and pressure to eat) as parents’ responses to child appetitive traits, which then influence child body composition (parent perceptions of child weight, BMI, and body fat). This conceptualization is consistent with viewing parental controlling feeding practices as modifiable behaviors that could be a point of target for prevention and intervention efforts to reduce pediatric obesity or childhood overweight [20,29].

A secondary objective of the present study was to explore the biopsychosocial process model of childhood eating and weight status across child sex, poverty status, and race/ethnicity. Research has found differential prevalence rates of pediatric overweight and obesity across sex, poverty status, and race/ethnicity, and differential pathways of susceptibility have been suggested across all three socio-demographic variables [32,33,34]. In particular, studies have shown that children of color (including Hispanic and black children) and children from lower income families exhibit greater risk for overweight or obesity [32,33,34].

## 2. Materials and Methods 

### 2.1. Participants

Participants are from the USA and consisted of 221 children between 4–6 years of age (51% males, (Mage) = 4.80 years, (SDage) = 0.85) and their parents (90.5% biological mothers, (Mage) = 32.02 years, (SDage) = 6.43) with 71% of the parents being married. Persons per household ranged from 2 to 10 (M = 4.25, SD = 1.43). Most parents (53%) reported a monthly household income of $3000 or below, and 10.5% reported a monthly household income above $9000. Based on persons per household and monthly household income data, families were classified as either above or at/below the poverty line using the U.S. Department of Human Health and Human Services guidelines. Among all the families, 36.6% were considered living at or below the poverty line. Children’s race/ethnicity was coded as White Non-Hispanic (48.4%), White Hispanic (28.5%), and Black (23.1%).

### 2.2. Procedures

This study was part of a larger study on child and parental factors associated with children’s emotional self-regulation, emotional eating, and body composition. Recruitment and data collection protocols for this study were approved by the university Institutional Review Board (ethic approval code: IRB2010-0496F). Recruitment and data collection protocols for this study were approved by the university Institutional Review Board (IRB). Participants were recruited from pediatricians’ offices, daycare centers and preschools, and local businesses that were commonly frequented by parents with young children. To increase the representation of ethnic and racial minorities in the study, we used a snowball sampling procedure so that parents who participated in the study referred friends and family who met the study criteria. Potential participants were provided with flyers with information on how to contact the researchers to participate in the study if parents had children between ages 4–6 years. Children and their parents were not eligible if (1) they were unable to use English fluently; (2) had a history of traumatic brain injury; (3) had a significant disability that would prevent them from completing the tasks in this study, such as blindness, etc. and(4) had food allergies related to the food groups (chocolate or grapes) that were provided in the larger study. Parents provided written informed consent before they and their children participated in the study. For the larger study, children and their parents visited the laboratory for one session to participate in a series of observational tasks not included in the present study, which lasted approximately 90 min. As part of the laboratory visit, parents completed a series of questionnaires on themselves and their child. Parents received $50 and children received a toy as a token of appreciation and compensation for their time and participation.

### 2.3. Measures

Primary study variables included child temperament (negative affectivity and effortful control), child appetitive traits (food avoidance and food approach), controlling parental child-feeding practices (restrictive feeding and pressure to eat), and child body composition. Parents provided information on child temperament, child appetitive traits, and parental child-feeding practices. Measures of child body composition were assessed using parent perceptions, BMI, and percent body fat.

Child temperament traits. Negative affectivity and effortful control were assessed as two temperament dimensions that are directly linked to emotionality and self-regulation. Parents rated children on negative affectivity and effortful control using items from the Child Behavior Questionnaire–Short Form (CBQ–SF) [35]. Negative affectivity was assessed with the anger, discomfort, fear, sadness, and soothability (reverse scored) subscales, and ratings were averaged across the 31 items (α = 0.72) to compute a score for negative affectivity. Effortful control was assessed with the attention focusing, inhibitory control, low-intensity pleasure, and perceptual sensitivity subscales, and ratings were averaged across the 26 items (α = 0.84) to compute a score for effortful control.

Child appetitive traits. Child appetitive traits (food approach and food avoidance) were reported on a five-point Likert scale (1 = never, 2 = rarely, 3 = sometimes, 4 = often, and 5 = always) by parents using the Children’s Eating Behavior Questionnaire (CEBQ) [20,36]. The food approach trait was assessed by the food responsiveness, enjoyment of food, desire for drinks, and emotional overeating subscales, with ratings being averaged across the 16 items (α = 0.86) to compute a score for food approach. The food avoidance trait was assessed by the satiety responsiveness, slowness in eating, fussiness, and emotional undereating subscales, with ratings being averaged across the 19 items (α = 0.71) to compute a score for food avoidance.

Controlling parental child-feeding practices. Controlling parental child-feeding practices (restrictive feeding and pressure to eat) were reported by parents using subscales of the Child Feeding Questionnaire (CFQ) [37]. The pressure to eat subscale consisted of four items (α = 0.76) such as “My child should always eat all of the food in his/her bowl.” The restriction subscale consisted of 10 items (α = 0.81) such as “I have to be sure that my child does not eat too much of his/her favorite foods.”.

Child body composition. Measures of child body composition consisted of data from parent perceptions, BMI, and percent body fat. Parents rated their perceptions of their child’s weight (as markedly underweight, underweight, normal, overweight, or markedly overweight) using items from the Perceived Child Weight subscale (α = 0.85) of the CFQ [37]. In addition, child weight and height were assessed by an experimenter, and child BMI was calculated using the U.S. Centers for Disease Control (CDC) gender- and age-relevant charts. The percent of child body fat was calculated using data output from a body composition machine calibrated for children (Tanita SC-331S comprehensive full body composition machine manufactured by Tanita Corporation of America, Inc., Arlington Heights, IL, USA). Over 20% body fat is typically considered overweight and below 13% body fat is typically considered underweight for 4–6-year-olds [38].

### 2.4. Data Analysis

Descriptive statistics and correlational analyses were performed using IBM SPSS Statistics 22 program copyrighted by IBM Corporation, Armonk, NY, USA. Structural equation modeling (SEM) was conducted to examine the relations between the major study variables by using Mplus version 7.2 [39] program copyrighted by Muthén & Muthén, Los Angeles, CA, USA. The hypothesized biopsychosocial process model is shown in Figure 1, with the effects of child temperament (effortful control and negative affectivity) on indices of child weight mediated by child appetitive traits (food avoidance and food approach) and controlling parental child-feeding practices (restrictive feeding and pressure to eat). The maximum likelihood method was used to estimate all path coefficients, which is widely used under the assumption of multivariate normality for the measured variables. In addition, while there were missing data, the missing rates were relatively small (4.65% for child BMI and child body fat, and 0.45% for the rest of the variables), which was lower than cutoff (i.e., 5% or 10%) and not problematic for this study [40,41]. The bootstrap confidence interval method was employed to test mediation effects. This method is recommended when analyzing a small sample size and a small mediation effect [42,43]. The analysis command of CINTERVAL (bcbootstrap) was applied to calculate bias-corrected confidence intervals in order to take into account bias and have more accurate estimation of the effect [44].

Differences between sex, poverty status, and race/ethnicity groups were also examined. Potential group differences were first tested on major variables by ANOVA tests. Then, three sets of multi-group analyses were conducted to examine whether path coefficients differed across sex, poverty status, and race/ethnicity. For each set of multi-group analyses, comparisons were conducted between the baseline model, which allowed for the freely estimated path coefficients across groups, and the constrained models, which constrained one path across groups to be the same at a time. Chi square differences tests were conducted between the fully unconstrained and fully constrained models to determine whether model paths differed across sex, poverty status, or racial/ethnic groups.

## 3. Results

### 3.1. Descriptive Statistics

Data were first examined for normality. Skewness and kurtosis (see Table 1) did not show a serious bias to normal distribution [45]. Mahalanobis distance statistic was used to identify multivariate outliers that influence normality. No outliers were detected since the Mahalanobis distances were smaller than the critical value χ^2^(degrees of freedom, 0.001) [46,47].

Descriptive statistics and correlational analysis were then examined (see Table 1). As expected, negative affectivity and effortful control were inversely related to one another. Pressure to eat and restrictive feeding were positively related to one another. Perceived child weight, child BMI, and child percent body fat were all positively correlated with one another. However, food approach and food avoidance were unrelated to one another.

### 3.2. Path Model

As shown in Figure 1, the hypothesized structural model was tested. All factors in the model were observed with the exception of a latent factor for child objective weight status indicated by BMI and percent body fat. The global model fit was evaluated by Chi-square test, comparative fit index (CFI), root-mean square error of approximation (RMSEA), and standardized root mean square residual (SRMR). The model fit was defined as acceptable when fit indices met the following criteria: TLI and CFI ≥ 0.95, SRMR ≤ 0.08 and RMSEA ≤ 0.06 [48]. Fit indices showed excellent model fit to the data (χ2 (20) = 24.09, *p* > 0.05; CFI = 0.99, RMSEA = 0.03, SRMR = 0.047). As shown in Figure 2, all path coefficients were significant with the exception of five paths that were not significant: the path coefficients between negative affectivity and food avoidance (β = 0.02, SE = 0.07, *p* = 0.72), effortful control and food approach (β = −0.12, SE = 0.06, *p* = 0.06), food avoidance and restrictive feeding (β = 0.08, SE = 0.06, *p* = 0.19), food approach and pressure to eat (β = 0.10, SE = 0.06, *p* = 0.11), and pressure to eat and perceived child weight (β = 0.03, SE = 0.06, *p* = 0.59). 

### 3.3. Mediation Analyses.

Analyses were conducted to test the hypothesized mediation effects, and three significant double or three-path mediation effects were found (see Table 2). First, the indirect effect of effortful control on child objective weight status was significant and mediated through food avoidance and pressure to eat (β = 0.06, *p* < 0.05, 95% CI = (0.01, 0.13)). Specifically, high effortful control is related to lower food avoidance, higher food avoidance is related to higher pressure to eat, and higher pressure to eat is related to low child objective weight status. Second, the indirect effect of negative affectivity on child objective weight status was significant and mediated through food approach and restrictive feeding (β = 0.09, *p* < 0.05, 95% CI = (0.03, 0.17)). Specifically, higher negative affectivity is related to higher food approach, higher food approach is related to higher restrictive feeding, and higher restrictive feeding is related to higher child objective weight status. Third, the indirect effect of negative affectivity on parent perceptions of child weight was significant and mediated through food approach and restrictive feeding (β = 0.1, *p* < 0.01, 95% CI = (0.05, 0.16)). Specifically, higher negative affectivity is related to higher food approach, higher food approach is related to higher restrictive feeding, and higher restrictive feeding is related to higher parent perceptions of child weight. Of interest is that the model accounted for 15.5% of the variance in perceived child weight and 8.5% in objective child weight status. 

### 3.4. Tests of Sex, Poverty Status and Racial/Ethnic Group Differences

Sex. There were no significant groups differences between girls and boys on almost all of the variables, with the only exception of girls (Mean = 5.39, Standard error = 0.06) scoring higher than boys (Mean = 5.15, SE = 0.06) on effortful control. As for the multi-group analysis, results indicate there were no sex differences in the model paths.

Poverty status. As shown in Table 3, participants classified as living in poverty scored higher on negative affectivity, pressure to eat, perceived child weight, child BMI, and child body fat, but lower on food avoidance, than participants not living in poverty. In addition, poverty status differences were found in the model paths. The Chi square differences test indicated that the path from pressure to eat to child objective weight status differed significantly between the group living above the poverty line and the group living at/below the poverty line (Δχ^2^(1) = 8.93, *p* < 0.05). The magnitude of path coefficient for the group living at/below the poverty line (β = −1.18, SE = 0.22, *p* < 0.01) was greater than the group living above the poverty line (β = −0.33, SE = 0.16, *p* < 0.05). Thus, results indicated that pressure to eat and child objective weight status had a negative relation in a way that children who had more pressure to eat tended to have higher scores on objective weight status. This negative relation was stronger in children from families living in poverty than children from families not living in poverty.

Race/ethnicity. Racial/ethnic differences were also found in the variables (see Table 4). The White Hispanic and Black groups scored higher than the White Non-Hispanic group on pressure to eat. The Black group scored higher than the White Non-Hispanic group on child body fat. The White Hispanic group scored higher than the White Non-Hispanic group on perceived child weight. 

Additionally, ethnic differences were found in model paths. The Chi square differences test indicated that two paths differed significantly across racial/ethnic groups: (1) racial/ethnic differences were found for the path from food approach to restrictive feeding (Δχ^2^(2) = 10.05, *p* < 0.05), and (2) racial/ethnic differences were found for the path from restrictive feeding to child objective weight status (Δχ^2^(2) = 6.54, *p* < 0.05). Further examinations of these two paths were then conducted. For the first path (i.e., food approach to restrictive feeding), Chi square differences test showed equal path coefficients between the White Non-Hispanic and White Hispanic groups, but they differed from the Black group (Δχ^2^(1) = 1.38, *p* > 0.05, CFI = 0.97, RMSEA = 0.055, SRMR = 0.09). The path coefficients for the White Non-Hispanic and White Hispanic groups (β = 0.89, SE = 0.13, *p* < 0.01) were greater than the Black group (β = 0.31, SE = 0.15, *p* < 0.05), indicating that White Non-Hispanic and White Hispanic children high on food approach were more likely to have parents who endorse restrictive feeding than Black children. For the second path (i.e., restrictive feeding to child objective weight status), Chi square differences test showed no difference between the White Non-Hispanic and White Hispanic groups, but they differed from the Black group (Δχ2(1) = 2.03, *p* > 0.05, CFI = 0.97, RMSEA = 0.056, SRMR = 0.10). The path coefficients for the White Non-Hispanic and White Hispanic groups (β = 0.33, SE = 0.14, *p* < 0.05) were smaller than those for the Black group (β = 1.31, SE = 0.04, *p* < 0.01). Thus, results indicated that restrictive feeding and child objective weight status had a positive relation in a way that children whose parents endorsed more restrictive feeding practices tended to have higher scores on objective weight status. This positive relation was stronger in Black children than Non-Hispanic and White Hispanic children.

## 4. Discussion

The primary purpose of the present study was to examine the pathways between early life child factors involved in appetite self-regulation (temperament and appetitive traits), controlling parental child-feeding practices (pressure to eat and restrictive feeding) and child body composition (Body Mass Index or BMI, percent body fat, and parental perception of child weight) in economically and ethnically diverse families. Study results are consistent with a biopsychosocial perspective on the development of children’s eating and weight [7] that links child biology and behavior with parenting practices, psychosocial processes, and environment such as poverty status and culture.

### 4.1. Relations Among Study Variables 

Study variables generally demonstrated construct validity based on results from correlations showing that measures of constructs that theoretically should be related, are in fact related. For measures of child temperament traits, negative affectivity and effortful control are two temperament dimensions that represent emotional self-regulation and correlations show that these dimensions are inversely related to one another, which is consistent with previous studies [49]. For child appetitive traits, reliabilities for food approach and food avoidance were adequate in our study (αs = 0.86 and 0.72, respectively) and food approach and food avoidance tendencies were orthogonal constructs, which is consistent with the previous validation data of the CEBQ showing that the majority, but not all, of the subscales for the food approach construct are negatively related to the subscales for the food avoidance construct [36]. Of particular interest to this study is that the two appetitive traits were related to parental child-feeding in ways that are consistent with prior research [15]. We found that food approach was positively correlated with restrictive child feeding, whereas food avoidance was positively correlated with pressure to eat. Thus, these measures of appetitive traits and their relations to measures of parental controlling feeding practices in the expected directions demonstrate predictive validity for food approach and food avoidance in our study. Also consistent with previous research [23] is that our measures of controlling parental child-feeding practices, pressure to eat and restrictive feeding, were positively related to one another. For our measures of child body composition, convergence was found across the three types of measures (parent perceptions, BMI, and percent body fat), with the three measures positively related to one another.

### 4.2. Links Between Temperament Traits and Appetitive Traits

Emotional self-regulation includes reactive or reflexive (e.g., bottom-up) as well as control or reflective (e.g., top-down) processes [50]. Negative affectivity or emotionality is considered a reactive or reflexive (bottom-up) dimension of emotional self-regulation. Prior studies have found a link between emotionality and slowness in eating and fussiness with foods [4,16], where children’s emotionality was assessed with the Emotionality, Activity, and Sociability Temperament Survey [51]. In the present study, we assessed children’s negative affectivity using the CBQ–SF, which included the anger, discomfort, fear, sadness, and soothability (reverse scored) subscales, and did not find a relation between negative affectivity and food avoidance. However, we conducted auxiliary analyses to examine relations between negative affectivity and the subscales of food avoidance (i.e., the satiety responsiveness, slowness in eating, fussiness, and emotional undereating) and those results show that negative affectivity was associated with fussiness with foods. Thus, the reactive/reflexive and bottom-up processes involved in emotional self-regulation (e.g., negative affectivity) are linked to some aspects of food avoidance (specifically fussiness with foods), which is consistent with prior research [4,16].

For effortful control, prior study findings suggest that measures of poor self-regulation are linked to food approach [4,16]. Somewhat consistent with prior research, effortful control was negatively correlated with food approach and the model path from effortful control to food approach was approaching significance in the present study. This suggests that the control or reflective (e.g., top-down) processes involved in emotional self-regulation (e.g., effortful control) are linked to appetite self-regulation or the ability to inhibit or to stop oneself from overindulging in desirable food or drinks [13,14].

### 4.3. Links between Appetitive Traits and Parental Controlling Feeding Practices 

Prior studies have found that children’s food avoidance is associated with maternal pressuring their children to eat, while children’s food approach and eating in the absence of hunger (particularly for girls) are associated with maternal restriction of unhealthy foods [15,25]. Consistent with prior research, food avoidance was related to pressuring children to eat but unrelated to restrictive feeding in this sample. Also consistent with prior study findings, food approach was related to restrictive feeding but unrelated to pressuring children to eat in this sample. These findings are from cross-sectional data, so we are unable to draw conclusions regarding potential bidirectional relations between child appetitive traits and parental child-feeding practices. However, we expect that parental child-feeding practices can also accentuate or attenuate the expression of child appetitive traits or appetite self-regulation [13,14], especially if parents are consistent with their parental child-feeding practices over extended periods of time. 

### 4.4. Links between Controlling Feeding Practices and Child Weight

Consistent with prior research [23,26,52], model results show that restrictive feeding was linked to child body composition (as indicated by parents’ perceptions and objective or anthropometric measures of child objective weight status). Pressuring children to eat was linked to low levels of child objective weight status but unrelated to parents’ perceptions of child weight. The negative relation between pressuring children to eat and child objective weight might be explained by parents who push their children to eat specific types of foods, such as those that parents consider nutritious or healthy including fruits and vegetables [23]. This scenario may particularly be likely for children who readily enjoy foods that are high in sugar, fat, or salt, as there is no need for parents to pressure or push their children to eat sugary, fatty, or savory foods. Furthermore, it is plausible that children who exhibit high effortful control may be able to better regulate their own eating so that their parents may feel less of a need to pressure or push their children to eat certain foods, resulting in relatively low rather than high child weight.

### 4.5. Mediating Mechanisms from Child Temperament to Child Body Composition

Model results show that child food avoidance and parental pressure to eat are two mediating mechanisms by which child effortful control has an influence on child objective weight status. Specifically, findings suggest that children with low levels of effortful control are more prone to exhibit food avoidance, which in turn is likely to elicit parental pressure to eat that in turn is linked to low child weight status. It is likely that children low on effortful control (i.e., executive attention and inhibitory control) tend to be very picky or fussy eaters, because effortful control reflects the ability to override or inhibit a reflexive or reactive (dominant) response in order to execute or activate an alternate (sub-dominant) response [49]. In other words, children who exhibit effortful control are less food avoidant or less picky and fussy about foods and, thus, more able to be receptive to parents presenting them with a variety of food choices even if they do not initially or naturally like such foods. When parents are offering children foods such as fruits and vegetables and “healthy” foods rather than foods that are nutrition-poor or excessively high in sugar and fat, pressuring or encouraging children to eat could then predict low child weight status [23]. Given that pressure to eat is negatively related to child objective weight status, children who are less receptive to parents presenting them with a variety of food choices (fruits, vegetables, and “healthy” foods) tend to be high on weight status. However, some parents may pressure their children to eat if they perceive their children to be underweight [53].

Model results also show that child food approach and restrictive feeding are two mediating mechanism by which child negative affectivity has an influence on “objective” or calculated child weight (i.e., BMI and percent body fat) as well as on perceived child weight; recall that model results accounted for 15.5% of the variance in perceived child weight and 8.5% in objective child weight status. Specifically, findings suggest that children with high levels of negative affectivity are prone to exhibit food approach, which in turn is likely to elicit restrictive feeding from parents that in turn is linked to high child objective weight status. Recall that negative affectivity refers to the disposition toward experience and expressing negative emotions (e.g., anger, discomfort, fear, sadness) without being able to be easily soothed. Children who are emotionally reactive and moody tend to exhibit food approach tendencies, including emotional overeating, which may then elicit restrictive feeding practices from parents and lead to high child weight status. Findings are consistent with prior research showing that both children’s food approach and parents’ restrictive feeding practices (particularly for weight-control purposes) are linked to higher weight status in children [20,26,27,28].

### 4.6. Sex, Poverty Status and Race/Ethnicity Differences

Several findings emerged across the demographic variables that are worth noting. First, the only sex difference that emerged was that girls scored higher than boys on effortful control, which is consistent with the literature on child temperament [54]. No sex differences were found in model results, so that the structural relations between emotional self-regulation, appetitive traits, parental controlling feeding practices, and child body composition appear to be the same across sexes. This is important as there has been research suggesting there may be differential risk factors and susceptibility among boys and girls pertaining to physical activity and social adversities [55,56]. For example, boys receive more benefit from physical activity than girls [56], whereas girls are more susceptible to social adversity between the ages of 1–3 years [55]. However, our findings suggest that relations between child temperament, child appetitive traits, and parental child-feeding practices do not differ across sexes in a sample of 4–6-year-olds. For childhood overweight and obesity prevention programs, this would imply that the information provided to parents on their feeding practices based on the child’s temperament and eating styles could work equally well for daughters and sons, at least in early childhood.

Prior research has documented that children of color (including Hispanic and black children) and children from lower income families are at heightened risk for overweight or obesity [32,33,34]. Consistent with prior research [57], we found that poverty status was associated with a host of major variables in the present study. Specifically, results show that families living in poverty had children with higher negative affectivity, lower food avoidance, and higher objective weight status, as well as higher parental pressure to eat and parental perception that their child was overweight. In addition, pressure to eat was associated with lower child weight in both groups, and this negative relation was stronger for families living in poverty than families not living in poverty. These findings are consistent with a biopsychosocial framework that highlights the confluence of biological and ecological or psychosocial risk or protective factors that jointly contribute to the risk or protection from the development of pediatric obesity. Further research is needed to examine how exposure to community or family stressors may influence parental practices that contribute to the family food environment in which children develop obesogenic eating habits and pediatric obesity. Findings from this study suggest parental child-feeding practices are dynamic and can be influenced by the child’s behavior as well as the family’s economic circumstances. 

Racial/ethnic differences were found in the mean-level of parental child-feeding, as well as in the structural relations between child appetitive traits and parental child-feeding practices, and between parental feeding practices and child objective weight status. Consistent with the literature [23], White Hispanic and Black parents reported higher levels of pressuring their children to eat than White Non-Hispanic parents. Results from structural equation modeling analyses indicate differential pathways across racial/ethnic groups between child appetitive traits and parental feeding and between parental feeding and child objective weight status. Specifically, in all ethnic groups, parents were more likely to endorse restrictive feeding for children who were high on food approach and this positive relation was stronger in White (Non-Hispanic and Hispanic) families than in Black families. In addition, in all ethnic groups, parents’ endorsement of restrictive feeding was associated highly with child objective weight status, and this positive relation was stronger in Black families than in White (Hispanic and Non-Hispanic) families. Our findings are consistent with research that suggests that behavioral interventions for pediatric obesity may need to take culture norms and ethnotheories of parenting and child growth into account, including adapting programs for Black families [58,59]. Thus, given differential effects across White (Non-Hispanic and Hispanic) and Black families, consideration of ethnic group-specific cultural values and beliefs and familial context on parental feeding and eating behaviors is important in the development and implementation of childhood obesity prevention programs.

## 5. Conclusions

The biopsychosocial process model in this study was based on the premise that child appetite self-regulation processes involve temperament and child appetitive traits that have biological bases, and these child processes could influence parents’ child-feeding styles and practices that then influence children’s eating and weight. This type of mediation chain (see Figure 2), in which one mediator transmits the influence of an independent variable to a dependent variable, has been referred to as a micromediational chain [60,61,62]. Our cross-sectional results are consistent with the biopsychosocial process model, but longitudinal data is needed to replicate findings to adequately address the causal influences or directionality of effects between child temperament, child appetitive traits, parental child-feeding, and child weight status. In addition, while child body composition was informed by three different types of measures (parents’ perceptions and objective or anthropometric measures of child weight status), all other constructs in the study were informed by parent-reports, albeit on reliable and validated measures.

In summary, our findings situate controlling parental child-feeding practices in the context of early life factors of child temperament and appetitive traits and a biopsychosocial framework of the development of appetite self-regulation and weight in childhood. Parenting practices are influential in the development of young children’s appetite self-regulation processes. Given that parental child-feeding practices are embedded in parents’ culture and tradition, they are not easy to change and would require interventions that take into consideration ways to address parental beliefs, attitudes, and perceptions regarding child health and foods [63]. Furthermore, our findings highlight that helping parents understand and learn how be responsive to their child’s temperament and appetitive styles in terms of parental child-feeding practices could be an essential component of promoting children’s development of appetite self-regulation and preventing childhood obesity [64].

## Figures and Tables

**Figure 1 nutrients-12-03353-f001:**
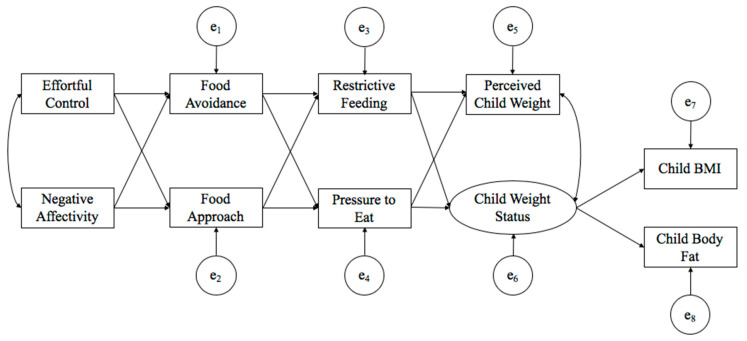
Hypothesized psychosocial process model of appetite self-regulation and weight status. Note. BMI = body mass index; e_1_ = error term for food avoidance; e_2_ = error term for food approach; e_3_ = error term for restrictive feeding; e_4_ = error term for pressure to eat; e_5_ = error term for perceived child weight; e_6_ = error term for child weight status; e_7_ = error term for child BMI; e_8_ = error term for child body fat.

**Figure 2 nutrients-12-03353-f002:**
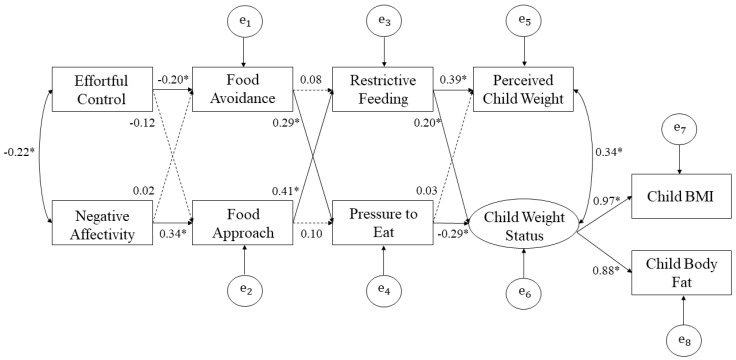
Final structural equation model with standardized path coefficients. * *p* < 0.05. Dashed lines are non-significant paths. Note. BMI = body mass index; e_1_ = error term for food avoidance; e_2_ = error term for food approach; e_3_ = error term for restrictive feeding; e_4_ = error term for pressure to eat; e_5_ = error term for perceived child weight; e_6_ = error term for child weight status; e_7_ = error term for child BMI; e_8_ = error term for child body fat.

**Table 1 nutrients-12-03353-t001:** Correlations and descriptive statistics for all variables.

	Variables	Mean	SD	Skewness	Kurtosis	1	2	3	4	5	6	7	8	9
1	Effortful Control	5.28	0.66	−0.36	−0.28	1								
2	Negative Affectivity	4.07	0.76	−0.01	0.04	−0.22 **	1							
3	Food Avoidance	1.35	0.60	0.08	0.98	−0.20 **	0.07	1						
4	Food Approach	2.49	0.60	0.25	0.12	−0.19 **	0.37 **	−0.01	1					
5	Restrictive Feeding	3.36	0.97	−0.45	−0.51	−0.10	0.25 **	0.08	0.41 **	1				
6	Pressure to Eat	2.58	1.10	0.24	−0.96	−0.07	0.13	0.29 **	0.10	0.14 *	1			
7	Perceived Child Weight	2.21	1.29	0.88	−0.41	−0.06	0.15 *	−0.01	0.28 **	0.39 **	0.09	1		
8	Child BMI	16.68	2.16	1.48	4.69	−0.12	0.05	−0.20 **	0.10	0.16 *	−0.25 **	0.34 **	1	
9	Child Body Fat	22.29	5.73	0.09	1.98	−0.10	0.05	−0.18 **	0.06	0.17 *	−0.21 **	0.31 **	0.85 **	1

Note: *, *p* < 0.05; **, *p* < 0.01.

**Table 2 nutrients-12-03353-t002:** The unstandardized indirect effects between temperament and weight. *Note*. SE = Standard error; CI = Confidence interval.

	Indirect Effect	Estimate	SE	95% CI
Effortful Control → Child Weight Status	Effortful Control → Food Avoidance → Restrictive Feeding → Child Weight Status	−0.01	0.01	(−0.04, 0.001)
	Effortful Control → Food Approach → Restrictive Feeding → Child Weight Status	−0.03	0.02	(−0.09, −0.01)
	Effortful Control → Food Avoidance → Pressure to Eat → Child Weight Status	0.06 *	0.02	(0.01, 0.13)
	Effortful Control → Food Approach → Pressure to Eat → Child Weight Status	0.01	0.01	(−0.001, 0.05)
Negative Affectivity → Child Weight Status	Negative Affectivity → Food Avoidance → Restrictive Feeding → Child Weight Status	0.001	0.003	(−0.004, 0.015)
	Negative Affectivity → Food Approach → Restrictive Feeding → Child Weight Status	0.09 *	0.03	(0.03, 0.17)
	Negative Affectivity → Food Avoidance → Pressure to Eat → Child Weight Status	−0.01	0.02	(−0.05, 0.03)
	Negative Affectivity → Food Approach → Pressure to Eat → Child Weight Status	−0.03	0.02	(−0.08, 0.01)
Effortful Control → Perceived Child Weight	Effortful Control → Food Avoidance → Restrictive Feeding → Perceived Child Weight	−0.01	0.01	(−0.04, 0.002)
	Effortful Control → Food Approach → Restrictive Feeding → Perceived Child Weight	−0.04	0.02	(−0.09, −0.001)
	Effortful Control → Food Avoidance → Pressure to Eat → Perceived Child Weight	−0.004	0.01	(−0.02, 0.01)
	Effortful Control → Food Approach → Pressure to Eat → Perceived Child Weight	−0.001	0.002	(−0.01, 0.001)
Negative Affectivity → Perceived Child Weight	Negative Affectivity → Food Avoidance → Restrictive Feeding → Perceived Child Weight	0.001	0.004	(−0.005, 0.02)
	Negative Affectivity → Food Approach → Restrictive Feeding → Perceived Child Weight	0.1 **	0.03	(0.05, 0.16)
	Negative Affectivity → Food Avoidance → Pressure to Eat → Perceived Child Weight	0	0.001	(−0.003, 0.01)
	Negative Affectivity → Food Approach → Pressure to Eat → Perceived Child Weight	0.002	0.004	(−0.004, 0.02)

Note: * *p* < 0.05; ** *p* < 0.01.

**Table 3 nutrients-12-03353-t003:** Poverty status differences for all the variables.

	Above the Poverty Line (*n* = 138)		Below the Poverty Line (*n* = 81)			
	Mean	SE	Mean	SE	F	*p*
Effortful Control	5.31	0.06	5.24	0.07	0.51	0.47
Negative Affectivity	3.96	0.06	4.21	0.08	5.88	0.02
Food Avoidance	1.41	0.05	1.26	0.07	3.24	0.07
Food Approach	2.49	0.04	2.45	0.08	0.19	0.67
Restrictive Feeding	3.41	0.08	3.27	0.11	0.99	0.32
Pressure to Eat	2.43	0.09	2.84	0.13	7.43	0.01
Perceived Child Weight	2.01	0.11	2.49	0.15	7.07	0.01
Child BMI	16.31	0.16	17.37	0.30	11.64	<0.01
Child Body Fat	21.58	0.41	23.68	0.84	6.41	0.01

Note: BMI = Body mass index; SE = Standard error; F = F statistic.

**Table 4 nutrients-12-03353-t004:** Racial/ethnic differences for all the variables.

	White Non-Hispanic (*n* = 106)		White Hispanic (*n* = 63)		Black (*n* = 51)			
	Mean	SE	Mean	SE	Mean	SE	F	*p*
Effortful Control	5.28	0.06	5.38	0.08	5.20	0.10	1.06	0.35
Negative Affectivity	4.04	0.06	4.02	0.11	4.19	0.12	0.84	0.43
Food Avoidance	1.46	0.06	1.38	0.07	1.14	0.07	4.99	0.01
Food Approach	2.53	0.05	2.46	0.08	2.43	0.10	0.62	0.54
Restrictive Feeding	3.33	0.09	3.36	0.15	3.44	0.11	0.23	0.80
Pressure to Eat	2.29	0.10	2.72	0.13	3.04	0.16	9.22	<0.01
Perceived Child Weight	1.90	0.11	2.58	0.18	2.31	0.18	6.17	<0.01
Child BMI	16.30	0.17	16.83	0.29	17.17	0.38	3.13	0.05
Child Body Fat	21.33	0.49	22.95	0.83	23.14	0.88	2.42	0.09

Note: BMI = Body mass index; SE = Standard error; F = F statistic.

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
