# Peer review of "Parental Child-Feeding in the Context of Child Temperament and Appetitive Traits: Evidence for a Biopsychosocial Process Model of Appetite Self-Regulation and Weight Status"

_nutrients, 2020, doi:10.3390/nu12113353_

Round 1

Reviewer 1 Report

This is a responsive revision. The clarification of the aims of the research and the emphasis on indirect paths between temperament, appetitive traits, feeding practices and child body composition sharpens the contribution considerably.

The clarification of the mediation analysis and the information from MacKinnon et al. was especially helpful. Similarly, the later reference to a micromediationnal chain. The inclusion of reference to top-down and bottom-up also strengthened the paper. Here the only reference you point a reader to is Bridgett et al. (2013). That paper is not about appetite self-regulation and it contains one brief reference to the top-down bottom-up approach. As this approach has recently been used in a number of publications about food-related self-regulation in children, it would probably help a reader not familiar with this approach if you included a couple of relevant references.

In section 3.3 where you summarize relationships between variables you use “low” and “high”. It would be more accurate here to use “lower” and “higher” throughout that paragraph.

I am still finding it a little difficult to master the role and idea of “emotion self-regulation” as you use it. In the abstract when you wrote about the main objective of the study you simply referred to the investigation of pathways from child temperament to body composition. The word or idea of emotion self-regulation did not appear in the abstract, and the abstract was clear and coherent without it. I looked at the title and wondered about whether or how the concept of emotion self-regulation added clarity to your work. I wondered whether it might work for you if the title was “Parental child-feeding in the context of child temperament and appetitive traits: Evidence for a biopsychosocial process model of childhood eating and weight status”.  Similarly, throughout the paper, I was not sure whether the term “emotion” was helpful. For example, on line 43 you refer to the temperamental basis for emotional self-regulation. Why refer here to only emotional self-regulation. Rather, it seems throughout the paper that you are arguing that temperament is related to self-regulation (not just emotion self-regulation). In particular, you refer to the two dimensions of temperament as contributing to bottom-up (negative affectivity) and top-down (effortful control) processes in self-regulation.   In the conclusion you also focus only on temperament and there is no mention of emotion self-regulation.

On line 64 you mention “effortful control representing control or reflective (e.g., top-down) processes.”  Here it might be more appropriate to use the term “regulatory” rather than “reflective”.

Line 496 there is a “to” missing.

Author Response

REVIEWER 1

Comments and Suggestions for Authors

The clarification of the mediation analysis and the information from MacKinnon et al. was especially helpful. Similarly, the later reference to a micromediationnal chain. The inclusion of reference to top-down and bottom-up also strengthened the paper. Here the only reference you point a reader to is Bridgett et al. (2013). That paper is not about appetite self-regulation and it contains one brief reference to the top-down bottom-up approach. As this approach has recently been used in a number of publications about food-related self-regulation in children, it would probably help a reader not familiar with this approach if you included a couple of relevant references.

  • We agree that the Bridgett et al. (2013) article is focused on emotional self-regulation rather than appetite self-regulation. Our measures are more directly relevant to child temperament (reactivity and self-regulation) and what we refer to as emotional self-regulation. However, we do appreciate Reviewer 1 calling our attention to the emerging work on the construct of “appetite self-regulation” which is very relevant to our study. We have revised section 4.2 to reference “appetite self-regulation”. We now reference work by Stoeckel et al. (2017) and Saltzman et al. (2018) to focus attention more specifically on potential relations between emotional self-regulation and appetite self-regulation. We believe these constructs are related but also somewhat different from one another, so we refer to both constructs in the manuscript and have revised multiple parts of the manuscript to reflect this nuance. As a result, we also modified the title of the manuscript to reflect the recognition of the construct of appetite self-regulation.

In section 3.3 where you summarize relationships between variables you use “low” and “high”. It would be more accurate here to use “lower” and “higher” throughout that paragraph.

  • Thank you for pointing out this detail and nuance and in Section 3.3. We have now revised this section.

I am still finding it a little difficult to master the role and idea of “emotion self-regulation” as you use it. In the abstract when you wrote about the main objective of the study you simply referred to the investigation of pathways from child temperament to body composition. The word or idea of emotion self-regulation did not appear in the abstract, and the abstract was clear and coherent without it. I looked at the title and wondered about whether or how the concept of emotion self-regulation added clarity to your work. I wondered whether it might work for you if the title was “Parental child-feeding in the context of child temperament and appetitive traits: Evidence for a biopsychosocial process model of childhood eating and weight status”.  Similarly, throughout the paper, I was not sure whether the term “emotion” was helpful. For example, on line 43 you refer to the temperamental basis for emotional self-regulation. Why refer here to only emotional self-regulation. Rather, it seems throughout the paper that you are arguing that temperament is related to self-regulation (not just emotion self-regulation). In particular, you refer to the two dimensions of temperament as contributing to bottom-up (negative affectivity) and top-down (effortful control) processes in self-regulation.   In the conclusion you also focus only on temperament and there is no mention of emotion self-regulation.

  • We really appreciate Reviewer 1’s comments about our construct of emotional self-regulation. As noted earlier in response to the point about appetite self-regulation, we have revised multiple parts of the manuscript to reflect the nuance between emotional self-regulation and appetite self-regulation, with the former referring to temperament-based self-regulation that is non-specific to foods and includes measures of negative affectivity (i.e., a measure of emotionality) and effortful control (i.e., a measure self-regulation). However, we now explicitly acknowledge the distinction between emotional self-regulation and appetite self-regulation. We believe that appetite self-regulation can be represented by the processes of both emotional self-regulation and appetitive traits. We have revised our manuscript to reflect this conceptualization. We appreciate Reviewer 1 helping us to clarify and refine our own thinking about these issues in this manuscript.

On line 64 you mention “effortful control representing control or reflective (e.g., top-down) processes.”  Here it might be more appropriate to use the term “regulatory” rather than “reflective”.

  • We have made this revision.

Line 496 there is a “to” missing.

  • We have made this corrections to the funding section.

Reviewer 2 Report

The authors have addressed all of my comments satisfactorily and I have no further points to add on the manuscript.

Author Response

We appreciate Reviewer 2's comments and feedback on our earlier draft, which really helped us improve our manuscript. We are glad that Reviewer 2 is satisfied with how we addressed earlier comments and concerns in our revision.

This manuscript is a resubmission of an earlier submission. The following is a list of the peer review reports and author responses from that submission.

Round 1

Reviewer 1 Report

Thank you for the opportunity to review this paper. The manuscript is well written and presents a more ambitious than usual statistical analysis of cross-sectional data. However, I have some serious concerns about conceptualization of some elements of the model as well as the interpretation of some of the findings. These, along with other minor comments and questions, are detailed below. I hope that the authors find these useful in either correcting or clarifying their study findings.

Specific comments

Page 2, line 61 – can you provide an example please?

Page 3, line 120 - add that the study participants are from the USA.

Page 4, line 138 – if participants had more than one child in the age range did they report on the oldest or youngest child?

Page 4, line 170 – has this approach with the CEBQ been used previously? If not, please provide a rationale. Also add the response options and scale anchors

Page 5, line 176 – why were these feeding practices (and the CFQ) selected and not others – such as emotional feeding etc?

Page 6, line 227 – the word error is misspelled

Page 6, line 241 onwards – here it would be good to describe whether the associations between the variables in these mediation models are positively or negatively associated with one another.

Page 10, line 341 - what about also looking at specific food approach scales - some .eg., enjoyment of food may not be as maladaptive as others e.g., emotional overeating.

combining the avoidance and approach scales may have its limitations!

Page 10, line 352 - what proportion of the sample were underweight though? why would parents get this advice if child not underweight? In other words - how many parents actually get this advice do you think and could that be enough to drive this association? Could it be that the heavier children are driving this relationship in that their parents are NOT pressuring them to eat?

Page 10, line 362 – in the model (Figure 2) pressure to eat is negatively associated with child weight status so I am confused as to why here you say it is associated with higher child weight status?

I think collapsing the food avoidance scales is proving problematic for your interpretation here. Some of these traits e.g., SE and SR are ones we might actually want to promote in children. So grouping these with FF I think is conceptually problematic and presents complications for your interpretation.

Page 10, line 371 – here you have said pressure to eat is negatively associated with child weight status but above (in this paragraph) you said higher weight status?

Perceived weight status should precede feeding practices in the model (see your point on page 12, line 445). The idea being that parents’ beliefs about their child’s weight (perceptions) precede their behaviours (feeding practices). This argument is the basis for the CFQ scales and has been presented frequently in the feeding practices literature. While the relationship with objective child weight status and feeding practices is likely bi-directional, it seems conceptually very difficult to argue that feeding practices would influence perceptions of child weight.

Page 11, line 401 – I think add here ‘at least for this age group’ (or something to that effect. Whether gender plays a role as children get older would need to be investigated.

Author Response

REVIEWER 1 Comments

Thank you for the opportunity to review this paper. The manuscript is well written and presents a more ambitious than usual statistical analysis of cross-sectional data. However, I have some serious concerns about conceptualization of some elements of the model as well as the interpretation of some of the findings. These, along with other minor comments and questions, are detailed below. I hope that the authors find these useful in either correcting or clarifying their study findings.

Specific comments

Page 2, line 61 – can you provide an example please?

*We now provide an example of effortful control on page 2, Lines 61 to 63: “For example, when a child who really loves chocolates gets a box of chocolates and is able to eat one piece, stop, and put the box away to save the rest to eat later, that child is exerting and demonstrating effortful control.”

Page 3, line 120 - add that the study participants are from the USA.

*We now mention that participants are from the USA on page 3, Line 122.

Page 4, line 138 – if participants had more than one child in the age range did they report on the oldest or youngest child?

*If parents had more than one child in the age range of the study, more than one child in the family were able to participate in the study as long as the child fit the study inclusion criteria. In such cases, parents needed to provide data for each specific child.

Page 4, line 170 – has this approach with the CEBQ been used previously? If not, please provide a rationale. Also add the response options and scale anchors

*This approach with the CEBQ has been previously used and published (e.g., Zhou, Liew, Yeh, & Perez, 2020). We now provide response options and scale anchors on page 4, Lines 168 to 170: “Child appetitive traits (food approach and food avoidance) were reported on a five-point Likert scale (1 = never, 2 = rarely, 3 = sometimes, 4 = often, and 5 = always) by parents using the Children’s Eating Behavior Questionnaire (CEBQ)”.  

Page 5, line 176 – why were these feeding practices (and the CFQ) selected and not others – such as emotional feeding etc?

*We focused on controlling parental child-feeding practices in this study, because a sizable body of research has shown that parental control is related to child effortful control. There is some, but less work on controlling parenting and child negative affectivity. However, there has been little to no work emotional feeding and child effortful control and negative affectivity. Thus, due to the complexity of the model we presented and tested, we decided to focus on controlling parental child-feeding practices instead of including the all the other types or styles of parenting and parental child-feeding practices are beyond the scope of this specific study.

Page 6, line 227 – the word error is misspelled

*Thank you. We have corrected this typo.

Page 6, line 241 onwards – here it would be good to describe whether the associations between the variables in these mediation models are positively or negatively associated with one another.

*We have added and clarified the direction of associations between the variables in the mediation models on page 6, lines 248 to 258.

Page 10, line 341 - what about also looking at specific food approach scales - some .eg., enjoyment of food may not be as maladaptive as others e.g., emotional overeating.

combining the avoidance and approach scales may have its limitations!

*We agree that some of the items within the food approach construct may not be as maladaptive as others. However, we did not tease apart of subscales within the food approach construct which consisted of 16 items with very good reliability among those scale items to represent the food approach factor (α = .86). We also fully agree with Reviewer 1 that combining the avoidance and approach scales have its limitations. Therefore, we distinguish these two constructs or factors in our study and in all of our analyses. Indeed, study results show different relations or pathways between child appetitive traits and child weight across the food avoidance and food approach factors. 

Page 10, line 352 - what proportion of the sample were underweight though? why would parents get this advice if child not underweight? In other words - how many parents actually get this advice do you think and could that be enough to drive this association? Could it be that the heavier children are driving this relationship in that their parents are NOT pressuring them to eat?

* We appreciate this feedback and have revised our interpretation for the finding of a negative relation between pressure to eat and child objective weight status on page 10, lines 364 to 371: “The negative relation between pressuring children to eat and child objective weight might be explained by parents who push their children to eat specific types of foods , such as those that parents consider nutritious or healthy including fruits and vegetables21. This scenario may particularly be likely for children who readily enjoy foods that are high in sugar, fat, or salt, as there is no need for parents to pressure or push their children to eat sugary, fatty, or savory foods. Furthermore, it is plausible that children who exhibit high effortful control may be able to better regulate their own eating so that their parents may feel less of a need to pressure or push their children to eat certain foods, resulting in relatively low rather than high child weight.”

Page 10, line 362 – in the model (Figure 2) pressure to eat is negatively associated with child weight status so I am confused as to why here you say it is associated with higher child weight status?

*Thank you for pointing out that typo. We have now corrected this.

 I think collapsing the food avoidance scales is proving problematic for your interpretation here. Some of these traits e.g., SE and SR are ones we might actually want to promote in children. So grouping these with FF I think is conceptually problematic and presents complications for your interpretation.

*We agree that some of the items or subscales may not be viewed as maladaptive or problematic, particularly when specific items or perhaps even a subscale is viewed individually (i.e., at the item- or even perhaps subscale-level). However, we are examining the variance that can be explained in child weight at the factor level based on the collection of 19 items that statistically show hang together well as indicated by good reliability for the food avoidance (α = .71). For this study, it would be appropriate to discuss findings based on the factor level rather than teasing out the individual subscales because that is not how the data and the path model was tested. Therefore, we were careful in the Discussion not to interpret findings based on the item or subscale levels to avoid confusion in readers that we are saying that satiety responsiveness or slowness in eating are maladaptive or problematic characteristics for child weight.

Page 10, line 371 – here you have said pressure to eat is negatively associated with child weight status but above (in this paragraph) you said higher weight status?

*Thank you for pointing out that typo. We have now corrected the typo that was pointed out above. Indeed, as Reviewer 1 noted, the relation between pressure to eat and child objective weight is a negative one.

Perceived weight status should precede feeding practices in the model (see your point on page 12, line 445). The idea being that parents’ beliefs about their child’s weight (perceptions) precede their behaviours (feeding practices). This argument is the basis for the CFQ scales and has been presented frequently in the feeding practices literature. While the relationship with objective child weight status and feeding practices is likely bi-directional, it seems conceptually very difficult to argue that feeding practices would influence perceptions of child weight.

*We have revised the interpretation of study results (please see above and page 10, lines 364 to 371). Our interpretation is consistent with the model results.

Page 11, line 401 – I think add here ‘at least for this age group’ (or something to that effect. Whether gender plays a role as children get older would need to be investigated.

*We have now added this caveat or clarification on page 11, line 418 that findings pertain to early childhood or our sample of 4- to 6-year-olds: “For childhood overweight and obesity prevention programs, this would imply that the information provided to parents on their feeding practices based on the child’s temperament and eating styles could work equally well for daughters and sons, at least in early childhood.”

Reviewer 2 Report

This research examines a number of important and relevant constructs related to outcomes such as weight gain/adiposity in children. The sample size is good, variables are measured in appropriate ways using validated parent-report procedures. The paper is clearly written. The findings have the potential to contribute to current knowledge about possible processes relating to children’s appetitive traits, parent feeding practices and child weight outcomes. The results for sex, poverty status and race/ethnic differences seem to be important. I wondered if the potential contribution could be enhanced by a reframing of some key ideas underlying the work, and some additional thoughts about the data analysis and results presentation.

One of the challenges in this work is that you want to focus on developmental processes, but the data are cross-sectional. The challenge with a cross-sectional design is to capture the extent to which or how the results can contribute to ideas and models about “pathways”. Correlations in the end are still correlations. Are you investigating “pathways” or associations? “Pathways” are used in longitudinal research in relation to processes or mechanism linking earlier measures to later outcomes. In longitudinal work it is easier to use causal reasoning. You use a lot of causal language throughout, especially in the discussion. Given the cross-sectional design, you probably should be less causal. The biopsychosocial model you draw on is longitudinal. In the title you suggest that you provide evidence for the model. Because your data are cross-sectional, you are not able to test aspects of the model. This does not stop you from framing the paper in terms of whether your cross-sectional data are consistent with the model. This can still be an important contribution to the field.

Processes in the biopsychosocial model you draw on are conceived as both transactional and bidirectional.  You say (line 105) that you acknowledge the bidirectional relationship between child eating behaviors and parent feeding practices but opted to assume a unidirectional relationship in the present work. There is also the possibility of bidirectional relationships between child weight and parent feeding behaviors. These bidirectional relationships have been extensively researched and it would help if you could included discussion of them. Certainly, there is scope for this in the discussion section. Your interest in possible processes should include consideration that many of these are bidirectional. The exclusive unidirectional emphasis at present is somewhat of a limitation of the paper.

Your model does not recognize that there could be direct paths between variables, such as between food approach and weight. Presenting these additional paths in the results model would be helpful. For example, there is a correlation between food approach and perceived weight but not between food approach and BMI or body fat. The opposite is the case for food avoidance, where there are correlations with BMI and body fat, but not for perceived weight. These finding are interesting and what might they tell us about process contributing to weight outcomes?

In Table 2 where you report tests of indirect effects, did you also examine direct effects? For example, did you consider the four steps of Baron and Kenny to establish mediation? It begins by first showing that the causal variable is significantly related to the outcome. For example, that negative affectivity is related to child weight status. From the individual correlations for BMI and body fat in this case, it looks like this is not so. The same for negative affectivity to weight status where there does not seem to be a direct path that is amenable to a mediation analysis. You did not explain how you did the mediation analysis. It would be important for a reader to understand what you did and how you have a significant mediation effect when there is no direct relationship between the temperament variables and the outcome variables.

The pair-wise correlations generally show little relationship between temperament and BMI and child body fat. You added together the individual subscales for negative affectivity and effortful control. In the case of EC, most of the emphasis in scholarship on self-regulation has been placed on inhibitory control. What was the relationships between inhibitory control and weight status (true, sometimes people find a relationship, and sometimes they do not). In the case of negative affectivity, again is there evidence that all of these subscales are linked to food approach, or is there evidence for some kinds of negative affect having a closer relationship with food approach?

Rather than going back to look at the individual components of negative affectivity and effortful control, you could somewhat rethink the biopsychosocial model. The foundation of a biopsychosocial model is indeed biologically-based child characteristics that set in train and interact with psychosocial processes. I am not sure that the model posits a direct relationship between temperament and weight outcomes. Rather, the idea is as you present it; temperament influences eating behaviors, which influences parent behaviors and some combination of these influence weight outcomes. Your results are clearly consistent with a biopsychosocial approach, but not necessarily in the mediation way that you examined.

You describe the main objective of the study as being to examine “potential mechanisms and pathways that might explain how child temperament is related to child body composition”. As I noted above, it seems that you are placing too much weight on a possible direct relationship between child temperament and weight outcomes. You could explain here what you mean by “is related to” child body composition. It seems that you are saying it is “related” through a number of processes. For example, that temperament could influence eating behaviors, which could influence feeding practices etc. But you would need to include here the possibility that some of this is likely to be bidirectional.

A core aspect of the research and its conceptualization concerns “emotion-related self-regulation”. You note that the two temperament factors examined, namely negative affectivity and effortful control are important for emotion-related self-regulation. You then link emotion-related self-regulation to food approach and food avoidance, with food approach and food avoidance as “expressions of approach and avoidance temperaments in relation to food, and thus are influenced by negative affectivity and effortful control” (lines 103-104). It is not clear here how negative affectivity and effortful control might influence food approach and food avoidance. There are some recent models of appetite self-regulation that could help you rethink the role of temperament. You explain the expected link between negative affectivity and food approach through processes such as emotional eating. You might find it helpful to draw on the “top-down, bottom-up” model (Russell & Russell IJBN&PA 2020; Bridgett et al., Psy Bull 2015) and/or the dual process model (Van Malderen et al. Appetite, 2020) of appetite self-regulation. Here effortful control is conceived as a top-down regulatory process and it would be expected that higher levels of this top-down control would be associated with regulation of both food avoidance and food approach tendencies. Food approach and food avoidance tendencies are both conceived as bottom-up or reactive responses to food. You found significant correlations between effortful control and both food approach and avoidance. Rather than drawing on “emotion-related self-regulation” as a core idea, a possibility is to emphasize bottom-up, top-down processes in appetite self-regulation. This would mean you are dealing with processes more closely related to eating behaviors and feeding practices than the somewhat indirect notion of “emotion-related self-regulation”.

Your discussion focused considerable on relationship between individual variables (sections 4.2 to 4.4). This seemed to diverge somewhat from the core purpose of the paper. As you note, there is an extensive literature about each of these relationships and it is not clear how much your research adds to that literature. Rather, your contribution is about how a biopsychosocial approach could help understand possible processes in the development of weight outcomes in children. There is plenty of scope in your results to set your discussion squarely on that matter, especially if you incorporate more of the top-down, bottom-up approach to appetite self-regulation. This would have to include speculation or consideration of bidirectional relationships.

Your results and discussion about sex, poverty status and race/ethnic differences are interesting and important. It would help if your introduction referred to these as part of the research purpose and provided some background. Here and also in the discussion it would be helpful if you could place sex, poverty status and race/ethnic differences in the context of possible differences in processes associated with an overall biopsychosocial approach.

In your conclusions you suggested that you “tested” the biopsychosocial model. That is a strong claim. It would be ok simply to say that the model “informed” the present research and you are investigating whether the cross-sectional results are consistent with the model.

What is the significance of the age of the children for your research and its key questions?

Minor comments

Line 35 “to” rather than “for”

Lines 56,57 could be clarified or expressed differently

Line 227 typo

If you have not seen it, the recent paper by Stifter and Moding in Appetite “Temperament in obesity-related research” could be helpful. They are also interested in temperament, eating behaviors, parent feeding and weight outcomes.

Author Response

REVIEWER 2 COMMENTS

Comments and Suggestions for Authors

Your model does not recognize that there could be direct paths between variables, such as between food approach and weight. Presenting these additional paths in the results model would be helpful. For example, there is a correlation between food approach and perceived weight but not between food approach and BMI or body fat. The opposite is the case for food avoidance, where there are correlations with BMI and body fat, but not for perceived weight. These finding are interesting and what might they tell us about process contributing to weight outcomes?

*We agree and acknowledge that there are likely direct relations between some of the variables that are not included or tested in the path model. We examined the direct relations between major variables using correlations that were included in Table 1. We did not include all possible direct paths in our path model, because the main goal was to examine the potential processes or mechanisms that underlie the indirect relation between child temperament (i.e., effortful control and negative affectivity) and child weight, and we considered child appetitive traits (food avoidance and food approach) and controlling forms of parental child-feeding practices (restrictive feeding and pressure to eat) as potential mechanisms that could help explain how child temperament is linked to child weight. Given a modest sample size and concurrent data, we tested the path model that we presented in this study without including all the potential direct paths to simplify the model and to focus in on testing the pathways that directly pertain to the primary aims of this study.

In Table 2 where you report tests of indirect effects, did you also examine direct effects? For example, did you consider the four steps of Baron and Kenny to establish mediation? It begins by first showing that the causal variable is significantly related to the outcome. For example, that negative affectivity is related to child weight status. From the individual correlations for BMI and body fat in this case, it looks like this is not so. The same for negative affectivity to weight status where there does not seem to be a direct path that is amenable to a mediation analysis. You did not explain how you did the mediation analysis. It would be important for a reader to understand what you did and how you have a significant mediation effect when there is no direct relationship between the temperament variables and the outcome variables.

*We used bootstrap confidence interval method to test statistical significance of mediated effects as we described in 2.4. Data Analysis. This method can be employed in Mplus and has been shown to have adequate statistical power and Type I error rates (MacKinnon, Lockwood, Hoffman, West, & Sheets, 2002). We did not use Barron and Kenny method which was shown to have low Type I error rates and low power (MacKinnon et al, 2002). Also, MacKinnon and colleagues (MacKinnon, Krull, & Lockwood, 2000) discussed that significant relation between the predictor and the outcome variable is not necessary to meaningfully interpret the relevant mediated effects as the requirement excludes the cases where the mediated effect and the direct effect have opposite signs and may cancel out. In this case, the total effect from the predictor to the outcome variable might not be statistically significant while there might be significant mediated effect.

  • MacKinnon, D. P., Lockwood, C. M., Hoffman, J. M., West, S. G., & Sheets, V. (2002). A comparison of methods to test mediation and other intervening variable effects. Psychological methods7(1), 83–104. https://doi.org/10.1037/1082-989x.7.1.83
  • MacKinnon DP, Krull JL, Lockwood CM. Equivalence of the mediation, confounding, and suppression effect. Prevention Science. 2000;1:173–181. 

The pair-wise correlations generally show little relationship between temperament and BMI and child body fat. You added together the individual subscales for negative affectivity and effortful control. In the case of EC, most of the emphasis in scholarship on self-regulation has been placed on inhibitory control. What was the relationships between inhibitory control and weight status (true, sometimes people find a relationship, and sometimes they do not). In the case of negative affectivity, again is there evidence that all of these subscales are linked to food approach, or is there evidence for some kinds of negative affect having a closer relationship with food approach?

*In this study, we focused on effortful control and negative affectivity as two broad dimensions of temperament that are most directly relevant for emotional self-regulation (often termed as emotion regulation in the literature). Inhibitory control is part of effortful control and ratings on the inhibitory control subscale was included in the computation of the effortful control composite.  In correlations, we did not find a significant correlation or direct relation between effortful control and weight status. In the case of negative affectivity, prior studies have found some evidence for the role of negative affectivity or proneness for negative affect and food approach, obesogenic eating behaviors, and emotional eating in young children (e.g., Hughes et al., 2008; Leung et al., 2014) and also in adults (e.g., Spoor, Bekker, Strien, & Heck, 2007).

Rather than going back to look at the individual components of negative affectivity and effortful control, you could somewhat rethink the biopsychosocial model. The foundation of a biopsychosocial model is indeed biologically-based child characteristics that set in train and interact with psychosocial processes. I am not sure that the model posits a direct relationship between temperament and weight outcomes. Rather, the idea is as you present it; temperament influences eating behaviors, which influences parent behaviors and some combination of these influence weight outcomes. Your results are clearly consistent with a biopsychosocial approach, but not necessarily in the mediation way that you examined.

*We agree with Reviewer 2 that our model does not posit a direct path or direct relation between temperament and weight outcomes. Indeed, as Reviewer 2 noted, our model is consistent with a biopsychosocial explanation of how children’s early biologically based characteristics such as their temperament partly drive their appetitive traits or eating styles. Parents’ child-feeding practices are partly in response to and also partly shaped by their children’s temperament and appetitive traits. The combination of child and parent factors contribute to children’s weight outcomes. Therefore, we agree with Reviewer 2’s interpretation of our findings which are also consistent with our conceptualization of these processes. If Reviewer read our model and interpretation of results differently, we want to make sure we can clarify it in the manuscript. But we are in agreement with Reviewer 2’s interpretation.

You describe the main objective of the study as being to examine “potential mechanisms and pathways that might explain how child temperament is related to child body composition”. As I noted above, it seems that you are placing too much weight on a possible direct relationship between child temperament and weight outcomes. You could explain here what you mean by “is related to” child body composition. It seems that you are saying it is “related” through a number of processes. For example, that temperament could influence eating behaviors, which could influence feeding practices etc. But you would need to include here the possibility that some of this is likely to be bidirectional.

*We appreciate Reviewer 2’s feedback on the need to more explicitly clarify the primary objective of this study. We have revised in the abstract and also in the text of the manuscript that “The main objective of this study was to examine potential mechanisms and pathways that might explain how child temperament is indirectly related to child body composition through child appetitive traits and parental child-feeding practices”. When we say related to, we include the possible of that relation being either direct or indirect. In the case of our study, we hypothesize that child temperament is indirectly related to child body composition. We hypothesize that there are mediating mechanisms between child temperament and child body composition, and our hypothesized model tests these mediating mechanisms that help explain the indirect relation between child temperament and child body composition. So we want to emphasize that we do not expect or hypothesize a direct relation between child temperament and child body composition. Rather, the relation is hypothesized to be indirect and through mechanisms such as child appetitive of food avoidance or food approach and parental child-feeding practices such as restrictive feeding or pressure to eat.

A core aspect of the research and its conceptualization concerns “emotion-related self-regulation”. You note that the two temperament factors examined, namely negative affectivity and effortful control are important for emotion-related self-regulation. You then link emotion-related self-regulation to food approach and food avoidance, with food approach and food avoidance as “expressions of approach and avoidance temperaments in relation to food, and thus are influenced by negative affectivity and effortful control” (lines 103-104). It is not clear here how negative affectivity and effortful control might influence food approach and food avoidance. There are some recent models of appetite self-regulation that could help you rethink the role of temperament. You explain the expected link between negative affectivity and food approach through processes such as emotional eating. You might find it helpful to draw on the “top-down, bottom-up” model (Russell & Russell IJBN&PA 2020; Bridgett et al., Psy Bull 2015) and/or the dual process model (Van Malderen et al. Appetite, 2020) of appetite self-regulation. Here effortful control is conceived as a top-down regulatory process and it would be expected that higher levels of this top-down control would be associated with regulation of both food avoidance and food approach tendencies. Food approach and food avoidance tendencies are both conceived as bottom-up or reactive responses to food. You found significant correlations between effortful control and both food approach and avoidance. Rather than drawing on “emotion-related self-regulation” as a core idea, a possibility is to emphasize bottom-up, top-down processes in appetite self-regulation. This would mean you are dealing with processes more closely related to eating behaviors and feeding practices than the somewhat indirect notion of “emotion-related self-regulation”.

*We appreciate Reviewer 2’s feedback on this issue. We have no changed the terminology from “emotion-related self-regulation” to “emotional self-regulation” throughout the manuscript. We also elaborated on page 2, lines 58 to 60 that “Negative affectivity and effortful control are two core temperament factors important for emotional self-regulation (also referred to as emotion regulation in the literature), with negative affectivity representing reactive or reflexive (e.g., bottom-up) processes and effortful control representing control or reflective (e.g., top-down) processes.”

Your discussion focused considerable on relationship between individual variables (sections 4.2 to 4.4). This seemed to diverge somewhat from the core purpose of the paper. As you note, there is an extensive literature about each of these relationships and it is not clear how much your research adds to that literature. Rather, your contribution is about how a biopsychosocial approach could help understand possible processes in the development of weight outcomes in children. There is plenty of scope in your results to set your discussion squarely on that matter, especially if you incorporate more of the top-down, bottom-up approach to appetite self-regulation. This would have to include speculation or consideration of bidirectional relationships.

*We have included brief discussion about top-down and bottom-up processes related to emotional self-regulation on page 10, lines 340 to 359. We also included brief discussion about the speculation or consideration of bidirectional relationships between child appetitive traits and parental child-feeding practices on page 10, lines 369 to 374: “These findings are from cross-sectional data, so we are unable to draw conclusions regarding potential bidirectional relations between child appetitive traits and parental child-feeding practices. However, we expect that parental child-feeding practices can also accentuate or attenuate the expression of child appetitive traits59, especially if parents are consistent with their parental child-feeding practices over extended periods of time.”

Your results and discussion about sex, poverty status and race/ethnic differences are interesting and important. It would help if your introduction referred to these as part of the research purpose and provided some background. Here and also in the discussion it would be helpful if you could place sex, poverty status and race/ethnic differences in the context of possible differences in processes associated with an overall biopsychosocial approach.

*We have emphasized and provided brief background on pages 3 and 4, line 122 to 128 that the examination of differences in pathways of the psychosocial process model of childhood eating and weight status across child sex, poverty status, and race/ethnicity as one of the objectives of this study. In addition, we elaborated in the discussion on page 12, lines 442 to 453 that “findings are consistent with a biopsychosocial framework that highlights the confluence of biological and ecological or psychosocial risk or protective factors that jointly contribute to the risk or protection from the development of pediatric obesity. Further research is needed to examine how exposure to community or family stressors may influence parental practices that contribute to the family food environment in which children develop obesogenic eating habits and pediatric obesity. Findings from this study suggest parental child-feeding practices are dynamic and can be influenced by the child’s behavior as well as the family’s economic circumstances.”  

In your conclusions you suggested that you “tested” the biopsychosocial model. That is a strong claim. It would be ok simply to say that the model “informed” the present research and you are investigating whether the cross-sectional results are consistent with the model.

*We appreciate this feedback and now realize that the language we used was inappropriate, so we now changed the sentence on page 12, line 453 to 456 as Reviewer 2 recommended.

What is the significance of the age of the children for your research and its key questions?

*We now added the rationale or the significance of the age of the sample for our research on page 2, lines 54 to 56: “We focus on this age group (4 to 6 years) for this study, because abilities for emotional self-regulation that involve executive or top-down control (including effortful control) begin to consolidate after the first 3 years of life.”

Minor comments

Line 35 “to” rather than “for”

*We have made this correction on line 35.

Lines 56,57 could be clarified or expressed differently

*We now realize that the sentence describing child temperament was poorly worded and confusing. We have revised the sentence on page 2, lines 56 to 58 to “Child temperament is biologically based10 and is considered the early foundation of later personality and long-term styles of behaving and reacting to the environment11”.

Line 227 typo

*We have corrected this typo.

If you have not seen it, the recent paper by Stifter and Moding in Appetite “Temperament in obesity-related research” could be helpful. They are also interested in temperament, eating behaviors, parent feeding and weight outcomes.

*We appreciate Reviewer 2 brining our attention to the Stifter and Moding (2019) article in Appetite. This article is extremely relevant article, and we have included and cited it in the study regarding interventions and considerations for future research.
